# Hydrothermal Liquefaction of Lignocellulosic and Protein-Containing Biomass: A Comprehensive Review

Abdul Sattar Jatoi [1], Ayaz Ali Shah [2,*], Jawad Ahmed [3], Shamimur Rehman [4], Syed Hasseb Sultan [5], Abdul Karim Shah [1], Aamir Raza [2], Nabisab Mujawar Mubarak [6], Zubair Hashmi [1], Muhammad Azam Usto [1] and Muhammad Murtaza [6]

1 Department of Chemical Engineering, Dawood University of Engineering and Technology Karachi, Karachi City 74800, Pakistan
2 Department of Energy and Environmental Engineering, Dawood University of Engineering and Technology Karachi, Karachi City 74800, Pakistan
3 Changzhou DUBO Polymer Co., Ltd., Changzhou 213002, China
4 Department of Computer System Engineering, Dawood University of Engineering and Technology Karachi, Karachi City 74800, Pakistan
5 Department of Chemical Engineering, Balochistan University of Information Technology Engineering and Management Sciences, Quetta 87300, Pakistan
6 Petroleum and Chemical Engineering, Faculty of Engineering, Universiti Teknologi Brunei, Bandar Seri Begawan BE1410, Brunei
* Correspondence: aas@duet.edu.pk; Tel.: +92-333-277-5895

**Abstract:** Hydrothermal liquefaction (HTL) is a thermochemical depolymerization technology, also known as hydrous pyrolysis, that transforms wet biomass into biocrude and valuable chemicals at a moderate temperature (usually 200–400 °C) and high pressure (typically 10–25 MPa). In HTL, water acts as a key reactant in HTL activities. Several properties of water are substantially altered as the reaction state gets closer to the critical point of water, which can result in quick, uniform, and effective reactions. The current review covers the HTL of various feedstocks, especially lignocellulosic and high protein-containing feeds with their in-depth information of the chemical reaction mechanisms involved in the HTL. Further, this review gives insight and knowledge about the influencing factors such as biomass pretreatment, process mode, process conditions, etc., which could affect the efficiency of the hydrothermal process and biocrude productivity. In addition, the latest trends, and emerging challenges to HTL are discussed with suitable recommendations.

**Keywords:** HTL; lignocellulosic biomass; protein-containing biomass; biocrude

## 1. Introduction

Lignocellulose is a viable feedstock among the abundant, sustainable, and ecologically benign resources that have received much attention in the effort to replace fossil fuels. The following benefits of lignocellulose, which includes wood, agricultural waste, and forestry waste, are superior to conventional fossil energy sources: (1) ample storage; (2) widespread distribution; and (3) low sulfur and nitrogen levels, which reduce processing-related pollution. Lignocellulose is a complex blend of cellulose (40–50%), hemicellulose (20–35%), lignin (20–40%), and other components [1]. The conversion is very challenging because the specific composition mostly depends on the species and the components are mostly connected by ethers, esters, hydrogen, and C-C bonds. A useful technology for turning it into liquid fuels and valuable compounds is hydrothermal liquefaction [2]. The intricacy of lignocellulose's structure makes its processing very difficult to manage, and therefore its mechanism of hydrothermal liquefaction has remained unknown [3,4]. The three primary parts of lignocellulose (cellulose, hemicellulose, and lignin) have been thoroughly investigated and discussed to comprehend the intricate mechanism [5,6]. Value-added applications of liquefaction products can be accomplished by developing a deeper

understanding of the liquefaction reaction mechanism [5]. As a result of a sequence of intricate processes, the macromolecular portion of lignocellulose in HTL produces biocrude, aqueous phase, gas phase, and solid residues. Since HTL is based on various feedstock sources, it has clear advantages over other conversion processes. It can process variety of feedstocks with a high-water content by avoiding drying process. Compared to pyrolysis products, HTL biocrude has a higher calorific value, reduced oxygen and moisture content, and superior stability. Biocrude produced through the HTL has a substantially higher energy density than biocrude produced through pyrolysis.

Additionally, the reaction parameters (temperature, pressure, heating rate, reaction time, solid-liquid ratio, and catalyst) primarily impact the yield and composition of HTL biocrude, etc. High-quality biocrude can be generated economically and sustainably by adjusting the reaction conditions [6,7]. Since 2015, more than 100 publications have been published on the liquefaction of lignocellulose or its components annually, with more than a quarter of these studies being HTL-related, according to the Web of Science database. One of the hotspots for research is the fundamental study of the HTL of lignocellulose. Currently, the United States, China, Canada, and other countries are the main locations for HTL research [8]. Similarly, high nitrogen-containing or proteinaceous biomass such as microalgae [9–11], sewage sludge [12], and food wastes [13] have also been used throughout the years in HTL and proved to be very efficient for bio crude production and other chemicals [14].

In the literature, there are several review articles that are already available [15–17]. However, a comprehensive information on different types of feedstocks, and their use in HTL under different process modes and parameters with the latest trends in HTL has not been compiled yet in a single review. A thorough examination of HTL of lignocellulosic and protein-containing biomass in terms of reaction-influencing variables, reaction mechanism, and product use can give readers a solid foundation in this area's theoretical underpinnings. Further, the key challenges and suitable recommendations related to HTL technology have been pointed down in the last part of the article.

## 2. Liquefaction Mechanism of Lignocellulose

The HTL process is particularly complicated since lignocellulose is a complex mixture of cellulose, hemicellulose, and lignin components (Table 1), each of which has a unique reaction mechanism [18,19]. Depolymerization of lignocellulose, decomposition of monomers, and recombination of reaction intermediates make up most of the HTL reaction steps [20,21]. Lignocellulose depolymerizes to produce monomers, breaking down into smaller fragments through decomposition processes such as cleavage, dehydration, and decarboxylation. These fragments undergo cyclization, condensation, and repolymerization to produce hydrothermal products such as biocrude, aqueous phase, gas, and solid residues [22,23]. It is challenging to develop precise reaction mechanisms and kinetic models for HTL due to the diversity of complex chemical processes that it entails and the abundance of reaction products they produce [22,24–26]. Numerous researchers have researched the liquefaction products and major lignocellulose component-degrading pathways in depth to better understand the HTL process.

**Table 1.** Chemical composition of biomass for HTL.

| Biomass | Lignin (%) | Hemicellulose (%) | Cellulose (%) | Ref. |
|---|---|---|---|---|
| Nut shells | 30–40 | 25–30 | 25–30 | [27] |
| Rice husk | 26–31 | 18–21 | 25–35 | [28] |
| Corn stover | 7–19 | 24–26 | 38–40 | [29] |
| Coffee husk | 9 | 7 | 43 | [30] |
| Sugar cane (bagasse) | 20–42 | 19–25 | 42–48 | [31] |
| Leaves and grass | 43.8 | 10.5 | 15.3 | [32] |

**Table 1.** *Cont.*

| Biomass | Lignin (%) | Hemicellulose (%) | Cellulose (%) | Ref. |
|---|---|---|---|---|
| Wheat straw | 17–19 | 26–32 | 33–38 | [29] |
| Coffee husk | 9 | 7 | 43 | [30] |
| Bamboo | 21–31 | 15–26 | 26–43 | [33] |
| Coir | 41–45 | 0.15–0.25 | 36–43 | [29] |
| Corn cob | 14–15 | 35–39 | 42–45 | [34] |
| Banana waste | 14 | 14.8 | 13.2 | [33] |
| Solid cattle manure | 2.7–5.7 | 1.6–1.33 | 16–4.7 | [33] |
| Corn fiber | 8.4 | 16.8 | 14.28 | [35] |
| Coir | 41–45 | 0.15–0.25 | 36–43 | [29] |
| Rice straw | 12–14 | 23–28 | 28–36 | [29] |
| Barley straw | 14–19 | 27–38 | 31–45 | [29] |
| Wheat straw | 17–19 | 26–32 | 33–38 | [29] |
| Pineapple leaf fiber | 5–1 | 18 | 70–82 | [36] |
| Sweet sorghum bagasse | 14–21 | 18–27 | 34–45 | [29] |
| Oat straw | 16–19 | 27–38 | 31–37 | [33] |
| Aspen | 19.5 | 21.7 | 52.7 | [37] |
| Eucalyptus | 26.9–28.2 | 12.7–14.4 | 46.6–50.3 | [38] |
| Japanese beech | 24 | 28.4 | 43.9 | [39] |
| Pine | 20 | 24–27 | 42–50 | [37] |
| Hardwood stem | 18–25 | 24–40 | 40–55 | [38] |
| Softwood stem | 25–35 | 25–35 | 45–50 | [39] |
| Paper | 0–15 | 0 | 85–99 | [28] |
| Cotton or seed hairs | 0 | 5–20 | 80–85 | [33] |
| Newspaper | 18–30 | 25–40 | 40–55 | [32] |
| Solid waste water | — | — | 8–15 | [30] |
| Waste paper from chemical pulp | 5–10 | 10–20 | 60–70 | [33] |
| Solid cattle manure | 2.7–5.7 | 1.4–3.3 | 1.6–4.7 | [34] |
| Bermuda grass | 6.4 | 35.7 | 25 | [34] |
| Swine waste | — | 28 | 6.0 | [30] |

## 2.1. Hydrothermal Liquefaction of Cellulose

D-glucose units joined together by -1, 4-glycosidic linkages make up the polymer known as cellulose. When cellulose is subjected to the HTL process, it breaks down into oligosaccharides and monosaccharides. These monosaccharide's can be transformed at high temperatures into aldehydes, furan derivatives, and small-molecular acids [40,41]. Glucose, fructose, erythrose, levoglucosan, 5-hydroxymethylfurfural (5-HMF), ethanol aldehyde, glyceraldehyde, acetone aldehyde, dihydroxyacetone, and certain oligosaccharides are the main end products of cellulose HTL (e.g., cellobiose, cellotriose, and cellotetrose). HTL produces glucose monomers due to the reaction with water that breaks cellulose's intramolecular and intermolecular hydrogen bonds [42]. Under hydrothermal conditions, glucose goes through several significant reactions, such as the isomerization of glucose to fructose by keto-enol interconversion, dehydration to produce 1, 6-anhydroglucose, breakdown to aldehydes and ketones by retro-aldol condensation, and dehydration of the

interconversion intermediate and fructose to produce 5-HMF. The reaction conditions have the greatest impact on the HTL products [43]. Only glucose is created when the reaction takes place at 400 °C for a relatively short period. Additional retro-aldol condensation processes as the reaction period lengthen, producing substances including glycolaldehyde, pyruvaldehyde, and glyceraldehyde [6,44]. Buendia-Kandia et al. investigated how temperature affected various cellulose hydrolysis reaction pathways and discovered that a low temperature (180 °C) accelerated the production of oligosaccharides from cellulose. Additionally, because homogeneous hydrolysis occurs at a slow rate at low temperatures, it is possible to recover oligomers with high levels of polymerization in the liquid phase [45]. The secondary reaction produced erythrose and acids when the reaction was conducted at a high temperature (260 °C) for less than 20 min. Levoglucosan and glucose were encouraged to develop at the moderate temperature (220 °C). The cellulose HTL products were likewise impacted by the reaction solvent's pH. Under acidic circumstances, the primary liquefaction products are levulinic acid and 5-hydroxymethylfurfural [46]. Under normal circumstances, the products are transformed into carboxylic acids such as lactic and acetic acids. Because $H_2O$ undergoes self-dissociation at high temperatures, generating $H^+$ and $OH^-$, acidic and basic conversion products are created under neutral circumstances. Transition metal sulfates in catalysts can also alter the selectivity of glucose during HTL, which promotes the conversion of glucose to products including lactic acid, levulinic acid, and formic acid [47]. For instance, at high temperatures, $Cu^{2+}$ and $Fe^{3+}$ efficiently convert glucose into levulinic acid and formic acid, while $Zn^{2+}$ and $Ni^{2+}$ encourage the conversion of glucose into lactic acid.

*2.2. Hydrothermal Liquefaction of Hemicellulose*

Hetero-polysaccharide hemicellulose comprises monomers such as xylose, mannose, glucose, galactose, and arabinose, among others. Hexose and pentose, two important monosaccharides hydrolyzed from hemicellulose, can be further dehydrated to produce furfural and 5-HMF [48,49]. The following reaction pathways are involved in the hydrothermal breakdown of hemicellulose: (1) alcohols and organic acids are the main products of sugar cleavage, (2) alcohols and organic acids undergo secondary reactions such as dehydration, oxidation, isomerization, and self-lactonization, (3) sugars undergo deoxygenation, oxidation, and self-lactonization to produce macromolecular fragments, and (4) hemicellulose and lignin fragments undergo a small amount of esterification and condensation reactions [6,50]. In lignocellulose, xylose typically exists in pyranose, furanose, or open-ring forms. According to studies, pyranose-type xylose is converted mostly to furfural. In contrast, open-ring xylose is converted primarily to break down products such as lactic acid, formic acid, glyceraldehyde, and acetone aldehyde [51]. In subcritical and supercritical water, xylose mostly undergoes isomerization, dehydration, retro-aldol condensation, and gasification reactions. In contrast to formic acid and organic carbon, vaporized during the reactions, dihydroxyacetone, formaldehyde, and acetic acid are stable and do not decompose to yield gas phase products [52]. Xylose decomposes quickly, and Paksung and Matsumura (2015) found that the primary liquid intermediates include furfural, retro-aldol condensation products, and organic acids. Xylose was only observable during the process at temperatures below 300 °C. Depending on how temperature affects the reactions, they can be separated into ionic and free radical reactions. Furfural is the main liquid product under subcritical circumstances where ionic processes predominate in favor of xylose dehydration [53]. Under supercritical conditions, free radical reactions prevail, promoting retro-aldol condensation with large glycolaldehyde yields. Yoon et al. showed in a semi-continuous reaction system by kinetic analysis of tulip xylan hydrolysis at 180–220 °C that low temperature and brief processing durations may result in high xylan conversion and oligosaccharide yields (up to 95%). The liquefaction processes of the other monosaccharides that make up hemicellulose are comparable under hydrothermal conditions. For instance, ethanol aldehyde, glyceraldehyde, and furfural are the main byproducts of the liquefaction of glucose, fructose, mannose, and galactose. The characteristics of the

biocrude created by glucose and xylose reactions were the same, indicating that the two molecules underwent similar reactions.

### 2.3. Hydrothermal Liquefaction of Lignin

The three basic components of lignin are p-hydroxyphenyl (H), guaiacyl (G), and syringyl (S). Lignin is an amorphous aromatic polymer molecule with anisotropic properties that the association of benzene propane units generates through various connections (ether and carbon-carbon bonds) [54]. Lignin has a complicated structure and a wide range of molecular weight distribution compared to cellulose and hemicellulose. Lignin's HTL enables the synthesis of oligomers, dimers, and monomers, assisting in the transformation of lignin into different aromatic compounds [55]. According to research published in the literature on the impact of lignin species, catalysts, and other factors on the products of HTL, the yield and distribution of degradation products are mostly affected by lignin's structural alterations [56]. The key processes that cause lignin to degrade during HTL include hydrolysis, ether and carbon-carbon bond breakdown, methoxy on the benzene ring fracture and degradation, and alkylation of groups on the benzene ring. Under HTL conditions, lignin is hydrolyzed and cleaved through ether linkages (such as -O-4, -O-4, -O-, 4-O-5, etc.) and carbon-carbon bonds (such as -, -5, 5–5, etc.) to produce a variety of phenolic compounds. As the temperature rises, the products undergo further hydrolysis, demethoxylation, and alkylation, resulting in the condensation of aromatic oligomers and alkylated aromatic hydrocarbons. Additionally, significant volumes of coke are produced due to the repolymerization of lignin HTL in intermediaries [56]. To convert lignin efficiently during HTL, it is essential to avoid condensation. Due to the complexity of lignin's chemical structure, research mostly focused on phenolic model substances such as guaiacol catechol, vanillic acid, diphenyl ether, and benzyl phenyl ether. Barbier et al. investigated the hydrothermal conversion of three lignin model compounds, namely vanilla, monobenzone, and 2, 2'-bisphenol, which reflect the methoxyphenol unit, ether link, and carbon-carbon bond of lignin (2012). Although the aromatic ring was mostly unaffected by the hydrothermal process, it was shown that the ether bond among the linkage bonds was more easily broken than the carbon-carbon bond [57]. Competitive cleavage (hydrolysis of ether bonds) and condensation reactions are the lignin's reaction pathways during hydrothermal settings (aromatic ring alkylation of intermediate products). Under HTL circumstances, lignin depolymerization primarily occurs through breaking ether bonds, with the -O-4 bond type being the most unstable. Lui et al. used model compounds with substituents to clarify the reaction pathway of the -O-4 aryl ether bond under hydrothermal circumstances to better understand the reaction mechanism of lignin during the hydrothermal process. The findings demonstrated that the conversion rate of -O-4 bonds was greatly boosted by alkyl groups on the -carbon and methoxy on the benzene ring. As a result of catalytic hydrolysis and elimination of -O-4 aryl ether linkages using OH- and H3O+, the major derivatives of phenol and alkenyl benzene were produced. Anisole, diphenyl ether, and phenethyl phenyl ether were selected as model compounds for lignin methoxy, 4-O-5, and -O-4 links to explore the mechanism and pathway of lignin hydrothermal reactions. Diphenyl ether was discovered to be stable between 260 and 290 °C, whereas anisole and phenethyl phenyl ether degraded. The 4-O-5 bond is more stable, and the -O-4 link in the lignin ether bond was preferentially cleaved under hydrothermal conditions. The -O-4 and methoxy bonds broke less [58]. The hydrothermal breakdown of a -O-4 lignin dimer at 175 °C and various hydroxide and carbonate concentrations was investigated by Nagel and Zhang [59] in 2019. The outcomes demonstrated that mild alkaline circumstances could successfully cleave the -O-4 bond, with guaiacol being the predominant product and vanillin and acetovanille being the minor products. Pathways for the breakdown of -O-4 lignin dimers have also been discovered. Before being cleaved, -O-4 lignin primarily takes the form of quinone methide; in alkali, it transforms into stable vinyl ethers and very unstable homovanillin. The -O-4 model compounds in diaryl ethers under hydrothermal conditions undergo depolymerization at mild conditions of 140–300 °C and 5–80 min,

whereas the model compounds of -O-4 undergo conversion at temperatures above 340 °C and 240 °C. Mind the hydrolysis rate of -O-4 bond is faster than that of -O-4 bond. Methoxy functional groups on the aromatic ring have been shown to facilitate depolymerization by reducing the temperature and reaction time required to dissolve the -O-4 and -O-4 ether linkages. Since the degradation temperature for the -O-4 structure is lower than that of methoxy, the temperature can be altered during the decomposition of lignin to reduce secondary reactions.

The pretreatment of lignocellulosic can also positively impact biocrude enhancement, especially by removing lignin; later, this lignin can produce phenolic derivatives. The physical and biological pretreatment methods for lignocellulosic biomass are listed in following Tables (Tables 2 and 3).

**Table 2.** Pretreatment of lignocellulosic biomass for HTL.

| Pretreatment Methods | Advantages | Disadvantages | Comments | References |
|---|---|---|---|---|
| Alkali | • Hemicellulose and lignin removal <br> • Low inhibitor formation <br> • Increases accessible surface area | • More residence times <br> • Salts formed are not recoverable | If this pretreatment method linked with the mechanical process addressed the mentioned issues | [35,60] |
| Acid | • High glucose yield <br> • Solubilizes hemicellulose <br> • Can be carried at ambient temperature | • High cost of acid <br> • High cost of corrosion-resistant equipment. <br> • Formation of inhibitors | Recycling of acids inhibitors (e.g., acetic acids) can be converted to valuable products and combined with a steam explosion. Addressed the mentioned issues or adoption of this process with catalyst leads to enhance the activity and selectivity of the process. That leads to an optimum quantity of acid, making the process more economical | [38,61] |
| Green solvent | • No toxic products <br> • Solvents are recoverable <br> • Dissolve a wide range of biomass <br> • Mild Process Condition | • High cost <br> • Solvent recycle cost | The deep eutectic solvent can be easily prepared. DES could be prepared easily by different hydrogen bond acceptors (HBA) and hydrogen bond donors (HBD) based on different molar ratios | [38] |
| Steam | • Cost-effective <br> • High substrate loading | • Partial degradation of Hemicellulose <br> • Size reduction cost | The process linked with acidic and alkali leads to addressing the issues mentioned slightly | [62] |
| LHW [a] | • Separation of nearly pure hemicellulose from rest of feedstock <br> • No need for a catalyst <br> • Hydrolysis of hemicellulose | • High energy and water input | Combined liquid hot water with sodium carbonate-oxygen pretreatment | [63,64] |
| AFEX [b] | • Removes lignin to an extent <br> • Low formation of inhibitors | • Ammonia recovery cost <br> • Less effective for high lignin content biomass | LAT process overcome this issue | [60,61] |
| ARP [c] | • Removes the majority of lignin <br> • Leaves high cellulose content | • High energy cost and liquid loading | Recycling of liquids/solvents combined with co-solvent to lower energy demand | [37] |
| Ozonolysis | • No generation of toxic compounds | • High cost of ozone | Combined with other pretreatments to reduce ozone consumption, e.g., aqueous ammonia | [38] |
| Super critical fluid | • Cost-effective <br> • Low degradation of sugar <br> • Ability to penetrate the crystalline structure | • High-pressure requirement | Combined with microwave to reduce utility usage | [39] |
| Organosolv | Removal of lignin and hemicellulose, leaving a high-purity solid glucan-rich fraction after solid-liquid separation | • The cost of solvent makes the process expensive | Acid-catalyzed organosolv pretreatment address this issue with a solvent recovery route | [39] |

**Table 2.** *Cont.*

| Pretreatment Methods | Advantages | Disadvantages | Comments | References |
|---|---|---|---|---|
| Wet oxidation | Efficient removal of lignin with the low formation of inhibitors | High cost of oxygen and alkaline catalyst | Recovery of alkaline catalyst can make the process more feasible | [39] |
| Concentrated acid | High glucose yield, ambient temperature | High cost of acid, corrosion, and inhibitor formation | A recovery route must be adopted to make the process feasible; corrosion can be addressed by adopting alloying material for construction | [63,64] |
| Diluted acid | Less corrosion problem with less inhibitor formation | Generation of degradation products, low sugar concentration | Combined with a steam explosion. Addressed the mentioned issues or adoption of this process with catalyst leads to enhance the activity as well as selectivity of the process | [39] |

[a] Liquid hot water, [b] ammonia fiber explosion, [c] ammonia recycle percolation.

**Table 3.** Biological pretreatment techniques and their possible outcomes.

| Pretreatment | Advantage | Disadvantage | Comments |
|---|---|---|---|
| Micro-organisms, i.e., fungi and bacteria | Low energy, cost with non-chemical degradation efficient for lignin | High reaction time with loss in carbohydrates | In-shortage pretreatment of wet biomass providing year-long delignification |
| Ligninolytic enzymes | Selective degradation of minimal lignin toxins | High cost of extraction and purification | Recycle the enzyme to economize the process |
| Cellulosic enzymes, i.e., cellulases | Hydrolysis of $\beta$-1, 4 linkages in cellulose | High cost of extraction and purification | Recycle the enzyme to economize the process |
| Xylanases | Efficient hydrolysis of hemicellulose fraction becomes crucial, and supplementation of accessory enzymes increases hydrolysis yields and thereby reduces enzyme costs and dosages | High cost of extraction and purification | Recycle the enzyme to economize the process |

## 3. Protein-Containing Feed for Hydrothermal Liquefaction

Protein-rich feedstocks include biomass, which includes things such as algae and grains, and biowaste, which includes things such as manure, municipal sludge, and food processing waste. Table 4 summarizes the components (crude fat or lipid, protein, carbohydrate, and ash) and the main elements that make up the biochemical compositions of these feedstocks (C, H, N, and O) [26]. The hydrothermal liquefaction of these protein-containing feedstocks during the past 20 years has produced some positive results regarding biocrude yields and high heating values. The composition of wastes produced during food processing might vary greatly depending on the facility, such as a cheese factory or a slaughterhouse [65]. The yields of biocrude often increase with the quantity of crude fat in the feedstocks, but lignin typically causes char to form under HTL. Due to the rapid decomposition of crude fat and nonfibrous carbohydrates, the maximum HTL bi-ocrude biocrude yield was obtained with feedstocks containing high levels of crude fat and nonfibrous carbohydrates at relatively lower temperatures (250–300 °C) and shorter retention times (5–30 min). After 2 s at 300 °C, Kabyemela et al. noted a conversion of glucose of 55%. Triacylglycerides (TAGs) were easily hydrolyzed in hot compressed water without catalysts. Higher reaction temperatures (300–350 °C) and longer retention times (30–120 min) are needed for feedstocks containing more proteins or fibrous carbohydrates [66]. According to Sasaki et al., at around 350 °C and 25 MPa, the reaction rate of cellulose starts to accelerate. The decomposition of amino acids from bovine serum albumin was reported by Rogalinski et al. in subcritical water hydrolysis at 250–330 °C and 4–180 s retention time, with an almost total decomposition of all amino acids at 330 °C and 200 s reaction time.

From HTL of food processing waste, additional value-added compounds can be created in addition to biocrude. The fish meat was processed by Yoshida et al. between 240 and 350 °C for 5 to 30 min to create biocrude and aqueous products with value-added compounds such amino acids, lactic acids, and phosphoric acids. To create amino acids and glucosamine, Quitain et al. hydrothermally treated shrimp shells at 250–400 °C for 5–60 min. Turkey offal was turned into biocrude and fertilizer using a hydrothermal technique by Changing World Technologies. Animal manure has less crude fat and more crude proteins and carbs when compared to food manufacturing waste. As a result, the HTL reaction temperature is raised to produce the best biocrude yields. In a batch reactor system, HTL of swine manure has been reported at 240–350 °C under reaction times of 5–180 min. With 71% to 78% carbon, 7% to 14% oxygen, 8.9% to 9.4% hydrogen, and 3.9% to 4.6% nitrogen, the HTL bio crude made from swine dung was shown to have high heating values of 35 to 39 MJ/kg [23]. Different process gases (such as CO, $N_2$, and $CO_2$) were also examined; CO was shown to be a reducing gas that increases the output and quality of bio crude. According to a study, storage time altered the biochemical makeup of the manure but had no impact on the output of HTL bio crude oil. Human feces are a promising HTL feedstock, similar to swine manure (Table 4). Pathogens are destroyed by HTL of human feces, which provides an extra advantage. However, if utilized as a transportation fuel, HTL bio crude produced from pig manure and human waste needs to be upgraded since it contains higher nitrogen and oxygen levels than petroleum. Under hydrothermal circumstances, swine excrement and crude glycerol liquefied. However, because of its dominant esterification and the accompanying high oxygen concentration in the biocrude, the heating value of the resulting liquefied HTL bio crude oil was significantly reduced, falling from 36 to 25 MJ/kg [23]. For 500 mL of cattle dung slurry, HTL of the manure was tested in the presence of NaOH as catalyst and beginning process gas (air, $N_2$, CO, and $H_2$. The formation of gases and char/tar rather than bio crude was caused by the higher initial pressure, longer reaction time, and increased solid content of the cattle dung. A 15 min reaction time at 250–350 °C with carbon monoxide as the process gas was examined using dairy manure as the HTL feedstock. At 350 °C, the energy recovery was at its maximum (68%) as measured by the difference between the energy in bio crude oil and that in raw manure. Because of their greater photosynthetic efficiency and reduced farmland requirement, algae are regarded as one of the prospective HTL feedstocks for next-generation bioenergy. Microalgae, macroalgae, and mixed-culture algae—which may include both micro- and macroalgae—are the three types of algae (Table 4). As HTL feedstocks, high-lipid microalgae have been favored. Dote used HTL to convert Botryococcus braunii into biocrude, and the yield of the biocrude was larger than the initial biocrude content in the algal biomass, showing that nonlipid fraction in algae had a role in the creation of the biocrude under HTL [23]. Low-lipid, high-protein algae, such as *Chlorella* and *Spirulina*, often produce more biomass in stressful environments such as wastewater than high-lipid algae do. As a result, interest in HTL of low-lipid algae has grown during the past five years. To increase the yield of algal biocrude and/or increase its heating value, reaction times, temperatures, beginning pressures, biomass-to-water ratios, total feedstock volumes, heating rates, reaction modes, catalysts, and reaction solvents were among the HTL characteristics that were investigated. Among all the HTL parameters examined, reaction temperature and reaction time are recognized as the two major factors affecting the quantity and quality of biocrude (heating value and elemental compositions). Due to their rapid growth and abundance, macroalgae were also considered as HTL feedstocks. However, macroalgae typically include large concentrations of ligno-cellulose and ash, both of which negatively impact the efficiency of HTL conversion [67]. For HTL conversion, mixed-culture algae from wastewater environments garnered more interest. Intake of nutrients from wastewater rather than fertilizers based on petroleum can reduce the load of nutrients on algae growth when algal bioenergy generation and wastewater treatment are combined. In an HTL-centered energy paradigm designed to improve the environment, nutrients in wastewater from post-HTL can be reused up to 10 times (via modeling) to boost algae growth and the production of biocrude. In a different

study, wastewater algae were shown to be a suitable HTL feedstock [68,69]. The typical illustration of algal HTL is demonstrated in Figure 1. High-ash, low-lipid algal biomass from lakes was collected for HTL in addition to reducing eutrophication. Municipal sludge is another encouraging feedstock that has equivalent HTL biocrude heating value to that of algae and animal manure. According to one study, the SlurryCarb method can convert sewage sludge into a biofuel with a maximum energy density of 9500 Btu/lb (dry basis) that can be used in traditional combustion infrastructure with only 20% more air [70–74]. The biochemical contents of the harvested sludge can be significantly changed by pretreatment procedures for municipal sludge in a wastewater treatment plant, and this has an effect on the quantity and quality of the resulting HTL biocrude. Vardon et al. used HTL to transform anaerobic sludge from a wastewater treatment facility into biocrude. However, the output was significantly lower than other sludge [75,76]. A study gap in the HTL of municipal waste has been recognized as the need for systematic studies involving HTL of various sewage sludge types and process variables, including reaction temperature and reaction time.

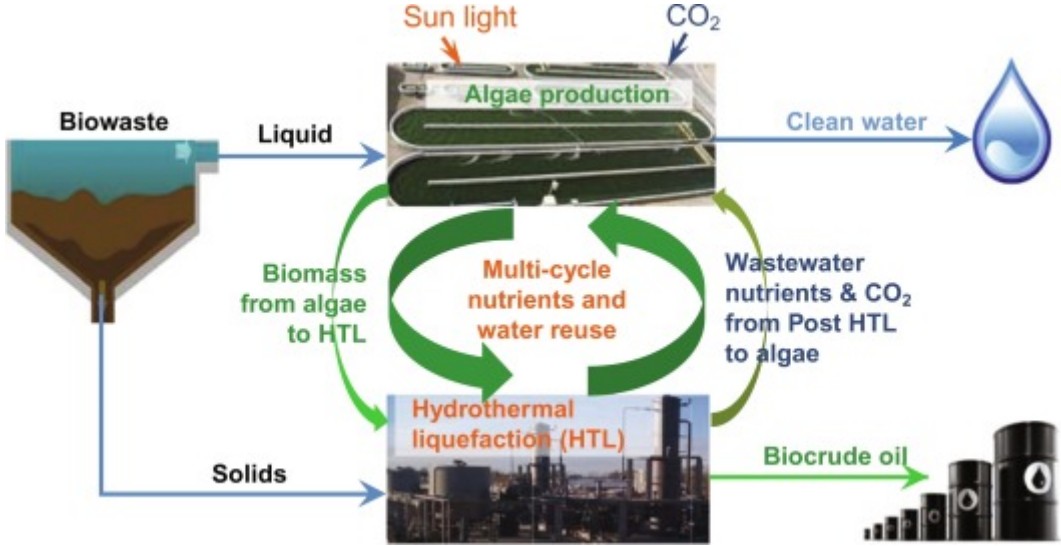

**Figure 1.** Process for a protein-containing feedstock for hydrothermal liquefaction [77].

**Table 4.** Summary of pretreatment methods for HTL feedstocks that contain protein contents.

| Feedstock | Value-Added Products during Pretreatment | Pretreatment Mechanism | Benefit | Limitation | References |
|---|---|---|---|---|---|
| Yeast (*C. curvatus*) | Polysaccharides and proteins derivatives | Low-temperature liquefaction (160–300 °C) | Produce value-added chemical and biocrude oil with lower N | Multistep processes, potential loss of organic matter | [78] |
| Microalgae (*Tetraselmis* sp.) | Protein derivatives | Low-temperature liquefaction (130–200 °C) | Produce biocrude with improved yields and lower N | Multistep processes, organic carbon lost to the pretreatment process water, high energy output | [79] |
| Microalgae (*Spirulina*, *Nannochloropsis*, *Chlorella*, and *Scenedesmus*) | Nitrogen-rich nutrient streams | Low-temperature liquefaction (125–225 °C) | Produce nutrient streams and biocrude oil with lower N | Multistep processes, organic matter lost to the pretreatment process | [80] |
| Mixed-culture algal biomass | N/A | Centrifugation and ultrasonication | Produce biocrude with improved yields and heating value | Multistep processes, require additional energy for pretreatment | [81] |

**Table 4.** *Cont.*

| Feedstock | Value-Added Products during Pretreatment | Pretreatment Mechanism | Benefit | Limitation | References |
|---|---|---|---|---|---|
| Microalgae (*Nannochloropsis*) | Biodiesel | Microwave irradiation | Produce biodiesel and biocrude simultaneously | Multistep processes, biocrude with lower HHV | [82] |
| Microalgae (*Nannochloropsis*, *Chlorogloeopsis*) | Lipids and phytochemicals | Microwave irradiation | Produce biocrude with lower N | Multistep processes, organic matter lost to the pretreatment process water | [83] |
| Microalgae (*Chlorella*) | Polysaccharides | Low-temperature liquefaction (140–200 °C) | Lower solid residue yield, produce biocrude oil with lower N | Multistep processes, loss of some organic matters | [84] |
| Swine manure | N/A | Filtration and centrifugation | Achieve a higher solid content of the feedstock, improve the energy recovery of HTL | Require additional energy for pretreatment | [85] |
| Microalgae (*Chlorella*) | Protein derivatives | Extracted with sulphuric or formic | Produce biocrude with lower N | Unselective removal of nitrogen-containing compounds | [86] |
| Microalgae (*Nannochloropsis* and *Scenedesmus*) | Crude lipids | Soxhlet extraction with hexane | Produce lipids and biocrude simultaneously | Produce biocrude with higher N, lower the yield of biocrude | [87] |
| Microalgae (*Scenedesmus*) | Crude lipids | Soxhlet extraction with hexane | Produce lipids and biocrude simultaneously | Produce biocrude with higher N, lower the yield of biocrude | [87] |
| Sewage sludge | Solid residue or biocharcontaining metals | Extracted with acetic acid, hydroxylammonium chloride, and hydrogen peroxide | Produce biocrude and biochar with lower metal concentrations | Required hazardous chemicals for demetalization, lower biocrude yield | [88] |

## 4. Factors Affecting the Hydrothermal Liquefaction

The reaction's process parameters mostly influence the distribution, composition, and characteristics of HTL products; therefore, optimizing those factors can aid in producing biocrude with a high yield and suitable quality [89]. To accomplish efficient biocrude production, HTL process parameters (such as lignocellulose type, reaction temperature, holding duration, heating rate, solid-to-liquid ratio, pressure, and catalyst) need to be considered and as discussed in the following sections.

### 4.1. Types of Lignocellulose

The yield and makeup of biocrude, the primary product, are highly correlated with those of lignocellulose [90]. In Figure 2, a few typical biomasses used for HTL are depicted. This is mostly because lignin has a more stable structure than other components, making it less likely to decay and liquefy to form biocrude. In contrast, cellulose and hemicellulose have simple structures, are less stable, and are, therefore, more likely to degrade and liquefy. Feng et al. [91] treated white birch, white spruce, and white pine bark with ethanol-water (50:50 *v/v*) at 300 °C for 15 min. The findings demonstrated that feedstocks with lower lignin concentration and higher ash content could boost conversion rates and biocrude yields. Liquidized white birch bark had the highest biocrude yield (67%), followed by white spruce bark (58%) and white pine bark's lowest (36%) levels. Due to lignin's higher thermal stability, its breakdown requires a higher temperature than cellulose and hemicellulose. Lignin is thermally more stable than the other biomass, and the order of hydrothermal conversion degree of biomass and biomass components was as follows: cellulose, sawdust, rice husk, and then lignin [92]. In the liquefaction of switchgrass in subcritical water, the residue solid mainly contained lignin fractions [93,94]. Softwood

biomass contains higher lignin content than hardwood biomass. It was reported that the lignin-rich cypress (softwood biomass) produced hydrocarbons with a major portion of phenolic hydrocarbons and derivatives than cherry (hardwood biomass), while the formation of acetic acid was more in the hemicellulose and cellulose-rich cherry than cypress [95]. In the hydrothermal conversion of the mixtures from different ratios of cellulose to lignin, the char yields increased with the increasing lignin content, and the yields of gas and aqueous soluble products increased with the increasing cellulose content, but it was difficult to conclude the biocrude yields change with increasing lignin content [96]. Belkheir et al. In this study, softwood Kraft lignin was depolymerized using subcritical water (623 K; 25 MPa) in a continuous, small pilot unit with a flow rate of 2 kg/h. $ZrO_2$, $K_2CO_3$/KOH, and $Na_2CO_3$/NaOH were used as catalytic systems and phenol as the capping agent. The influence of the ratio between sodium and potassium in the feed on the yield and composition of the product stream was investigated. The results showed that biocrude, water-soluble organics (WSO), and char yields were not remarkably influenced by shifting the catalytic system from potassium to sodium [97]. Yang et al. performed liquefaction test on five model components, including xylan (hemicellulose), crystalline cellulose, alkaline lignin, soya protein and soybean oil at 290 °C, and found the trend of biocrude yield was highest to lowest as: lipids, protein, cellulose, hemicellulose, and lignin [76]. The yields of biocrude from several lignocellulosic feedstocks under different HTL parameters are displayed in Table 5.

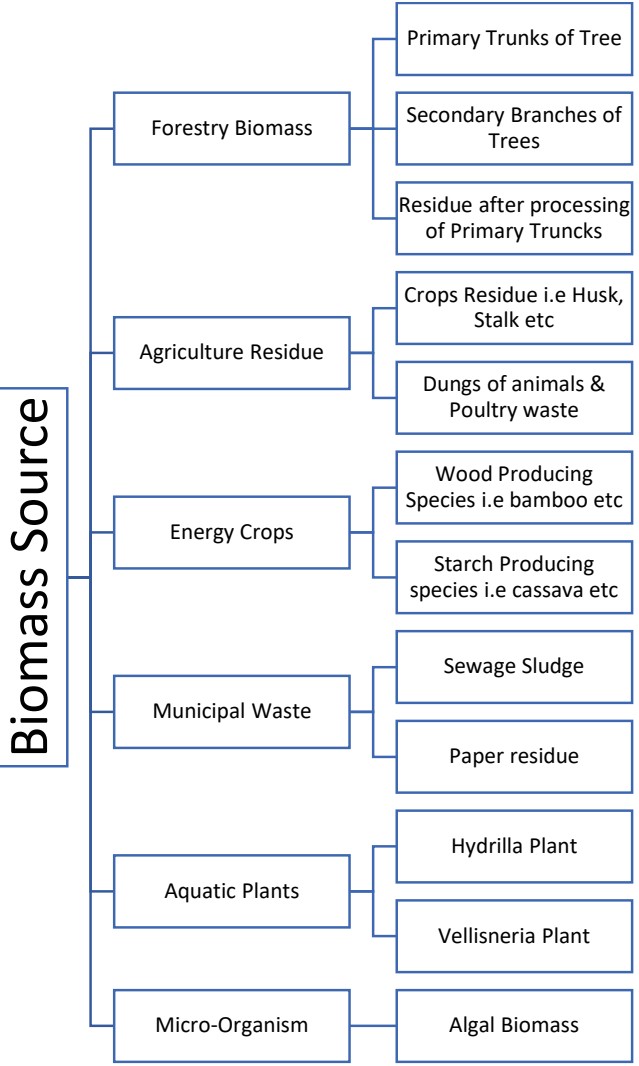

**Figure 2.** Different biomass for HTL.

*4.2. Process Mode*

Batch-type HTL reactors are most common in the literature because of their simple working mechanism [17]. In a batch-type reactor mixture of water, biomass, and catalyst (if needed) are loaded into an autoclave, where it is heated at a certain temperature and selected reaction time; then it is cooled down, and products are collected and analyzed. Any type of material can be screened in an autoclave at different process parameters and operating conditions [98,99]. Conversely, there are some disadvantages of using a batch reactor; for instance, **Thermal transience:** A batch-type reactor system has to go from ambient conditions to the desired temperature and pressure; hence, process conditions are not constant. This transience makes it difficult to separate the effects of temperature and time. **Difficulty in decoupling temperature and pressure:** In batch-type experiments, the pressure is dependent on the mixing of reactants; hence, the experimental conditions are often those as of the saturation conditions of the water. The Pre-pressurizing the system with an inert gas can partially overcome this problem [100]. In a continuous system, pressure and temperature can be controlled in a completely independent fashion. **Different contact pattern:** In a batch-type reactor, the feedstocks are mixed through by an impeller or shaking the reactor [101]. Whereas in a continuous reactor, it is mixed through a continuous stirred flow reactor (CSTR), new feedstocks are continuously supplied, and products are removed. **Significant distance toward actual industrial implementation**: The industrial application of a batch-type reactor is very limited due to its low production capacity of the products and high energy consumption.

Therefore, it is evident that batch processing alone is not able to provide results that can be directly utilized for the industrial development of the process. Moreover, testing in continuous devices allows for experiencing some technical issues and facts that are typical of continuous processing. One of these is, for example, high-pressure pumping, etc. There are several different studies in the literature that reveal that continuous HTL reactors are available in different sizes, from very small laboratory-scale plants to very large demonstration industrial-scale plants. For instance, the group of Elliot and coworkers at the Pacific Northwest National Laboratories (PNNL) built a continuous batch-type reactor to process different types of biomass with a special focus on the processing of algae macro, microalgae, and residues from agroindustry such as grape pomace. The process diagram is shown in Figure 3. The working mechanism of this plant is described as the slurry is placed in the two pressurized feed tanks by use of a syringe pump (a modified Isco 500D dual system), where the mixture is compressed at a pressure of 200 bar and preheated at 130 °C in a horizontal biocrude jacketed preheater then it is transported to a CSTR reactor, where the mixture reaches the reaction temperature. During this process, the volume of this reactor is altered from 1 L to 400 mL, and plug-flow reactor (PFR) is placed after that to increase the residence time. According to the authors, this alternative approach was adopted to minimize the plugging issues experienced with the PFR reactor, especially when operating with lignocellulosic feedstocks. This setup has also been used for many studies with some adaptations such as different types of algae such as *Nannochloropsis* sp. [102], macroalga *Saccharina* spp. [103], and *Chlorella*, high lipid content [104] and agricultural waste [105], such as grape pomace [106], and wastewater solids [107].

At the University of Sydney, Australia [107], algae, in particular, *Chlorella*, *Spirulina*, and *Oedogonium*, were experimented in a laboratory-based pilot scale plant with a range of 350 °C and 250 bar. All the experiments were conducted at subcritical conditions, with low values of dry matter content (mostly 1–5%, with a few attempts at 10%) and residence times of 3–5 min. The results revealed the maximum biocrude yield of 42% with 10% of dry matter content at a given temperature. At the University of Illinois at Urbana Champaign, USA, Ocfemia et al. [108] established and tested a pilot plant the reactor configuration is CSTR with a residence time of 60 min for the liquefaction of swine manure [108,109]. At Iowa State University, USA, by Suesse et al. [110], the temperature and pressure range of (450 °C and 690 bar) was designed by Supercritical Fluid Technologies Inc. (Newark, Delaware), where different waste streams, specially Rhizopus oligosporus at 270 bar was

processed and produced biocrude yields up to 48–61% with an oxygen content of 12–16%. It was also noted that switching from subcritical to supercritical conditions did not lead to considerable change in the yields, whereas the quality of different products reported that similar characteristics of microalgae biocrude were found at 300 °C.

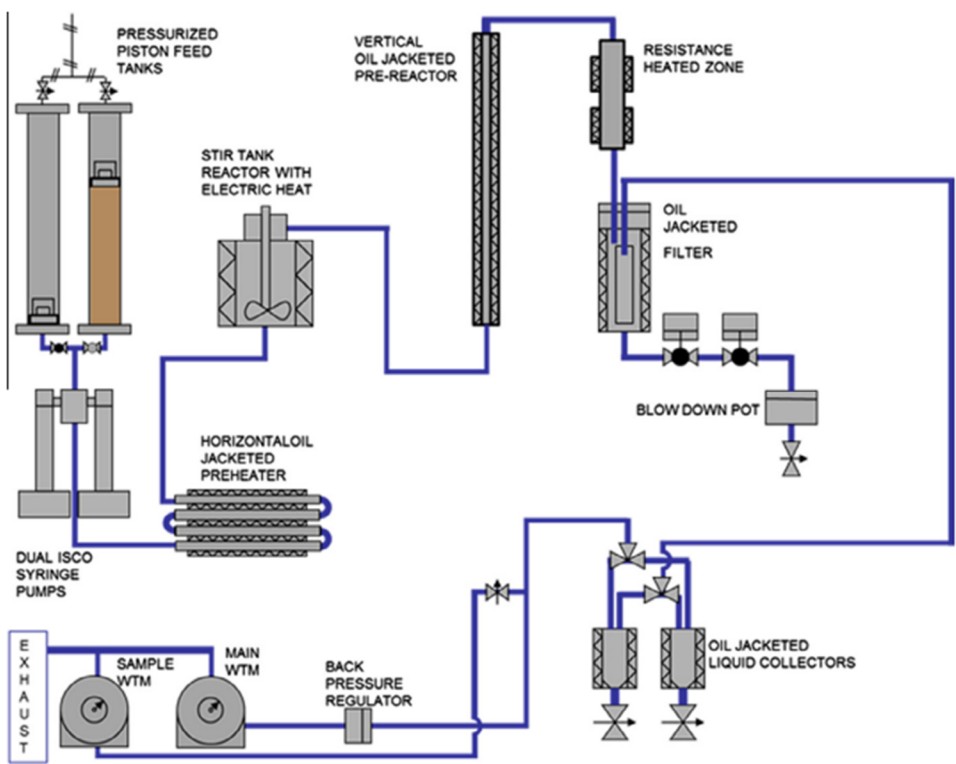

**Figure 3.** Process scheme of the plant in Pacific Northwest National Laboratories, adopted from [104].

The Kraft lignin (by-product of the pulping industry) has also been used by Chalmers University of Technology (Gothenburg, Sweden) with 0.5 L fixed bed reactor, manufactured at Inconel and packed with pellets of zirconia ($ZrO_2$) [111,112]. This experimental setup consists of a mixture composed by deionized water, lignin, $K_2CO_3$ and phenol (which was adopted as a char suppressing agent). The yields of biocrude ranging from 58% to 74%, although the produced biocrude presents a relatively high oxygen content (15–21%) compared to the feed (26%) [111,113].

Leng et al. stated that PFR often achieves better conversion yields than CSTR in a reactor of the same volume. When compared to plug-flow systems, CSTR systems have the inherent disadvantage of mixing mode, which means that some of the feedstock will always be subjected to unfavorable reaction time, some undercooked and some overcooked [114]. This disadvantage exists even though CSTR systems may be easier to operate and maintain, such as cleaning. It is challenging to compare batch and continuous HTL systems because the batch products are frequently equilibrium constrained. Although the yields and heating values of the biocrude produced from swine manure and algae are comparable in continuous modes, the ideal reaction conditions for these two feedstocks in a continuous HTL reactor are very different [115].

It is commonly accepted that for algal biomass, harsher reaction conditions can result in larger yields, lower oxygen content, and higher nitrogen content in the HTL biocrude. Additionally, algal biocrude yields from extremely short residence times in continuous reactors (such as 1–5 min) might be comparable to those from longer residence times in batch modes (such as 60–120 min). Recent batch and continuous HTL investigations also imply that the synthesis of biocrude is favored by faster heating rates and shorter residence times [116].

Through a continuous HTL, this characteristic would also help to increase the energy efficiency and technological viability of biomass-to-fuel systems. Compared to algal feedstocks, continuous HTL of swine dung necessitates a longer retention period. Ocfemia et al. [108,109] performed a 40–80 min continuous HTL of swine dung with a CSTR at 285–325 °C. The best biocrude yield was obtained between 285 and 305 °C for a retention time of 60 min, which is quite close to the results reported in a batch reactor.

According to this continuous HTL investigation, increasing the reaction temperature from 305 °C to 325 °C lowered the biocrude production by 22–25 wt.% at the same residence time, whereas increasing the residence time from 60 to 80 min only increased the biocrude output by 1–2 wt.% at 285–325 °C. The literature states that swine manure has a protein content of 25 wt.% and a carbohydrate content of 35 wt.%, whereas the algal feedstocks used in these continuous HTL studies have a much higher protein content (for example, 60–68 wt.% in *Chlorella* and *Spirulina*). Additionally, whereas the continuous HTL of swine manure has been studied in a CSTR, the continuous HTL of algal biomass has been studied in a PFR or in conjunction with a CSTR and PFR [117].

Some continuous bench-scale units are also available. The bench-scale plants are mostly utilized at universities and research institutions for fundamental research or as first experimental devices to obtain data in view of a future scale-up. They are normally of reduced sizes, with reactor volumes often no larger than a few hundred milliliters. One of the first documented studies was carried out at the Karlsruhe Institute of Technology, Germany. Here, a small continuous device was built for the HTL of baker's yeast and other residual biomass [118–120].

A continuous bench-scale unit (CBS1) was established at Aalborg University (Denmark) in collaboration with the Steeper Energy ApS Company. The features of the CBS1 plant are a 10 L tubular reactor designed to operate at supercritical conditions, and an important aspect of this plant is to use of water as recirculating solvent [121,122]. This water phase recirculation enhances the biocrude yield as well as its hydrogen–carbon ratio [100]. In this same study, the author investigated the utilization of glycerol as a co-solvent to reduce the formation of char and to process high organic content.

A continuous HTL pilot plant at Aarhus University (Denmark) has built and commissioned with capacity of around 1 L/min. This HTL plant contains a high-pressure pump to drive the biomass slurry up to 220 bar with maximum temperature of 350 °C, it has also the hydraulic oscillator to increase turbulence in reactor system by achieving better mixing, uniform residence time and to enhance better heat transfer [71]. For better understanding about the all the processes concerning with continuous HTL, the Castello et al. presented a detailed review covering all possible technical and financial aspects in view of existing and future scope of continuous HTL [123].

### 4.3. Process Conditions

Process parameters for HTL include reaction temperature, reaction time (also known as retention time), pressure, feedstock/water ratio (solid content), and catalyst applications. The distribution of HTL product yields generated from algal biomass is shown in Figure 4, which summarizes the influence of key process factors (reaction temperature, reaction duration, and total solid content of feedstocks) on that distribution. The most important element influencing HTL product yields and the quality of biocrude produced from feedstocks such as animal dung, algae, and food processing waste has been identified as reaction temperature. Reaction temperatures between 250 °C and 375 °C were typically used to generate biocrude from biomass-containing proteins [124]. The species of the feedstock has a significant impact on the reaction temperature. Microalgae (*Chlorella*) were converted into biocrude by Yu et al. and Gai et al. without the use of a catalyst, and they proposed that 280–300 °C is the ideal reaction temperature to maximize the yield, heating value, and energy recovery of the biocrude. Using HTL, Brown et al. and Valdez et al. transformed *Nannochloropsis* into biocrude and concluded that the optimal reaction temperature for achieving the highest biocrude output is between 300 °C and 350 °C. The ideal reaction

temperatures vary greatly depending on heating rates, mixing versus nonmixing, and reactor systems (such as batch versus continuous). Therefore, it is crucial to clarify the reaction mechanisms of HTL in terms of the makeup of the feedstock, the reaction temperature, the duration of the reaction, and the catalyst. According to numerous research, more gas products are produced when the HTL reaction temperature is higher than 320 °C. Li et al. discovered that raising the HTL reaction temperature from 320 °C to 380 °C significantly increased the gas product yields. The initial stage's controlling processes for feedstock are hydrolysis and depolymerization, followed by repolymerization between 220 °C and 375 °C and gasification above 375 °C. More solid residues and char would form as the reaction temperature rose over the reaction regime of repolymerization [74].

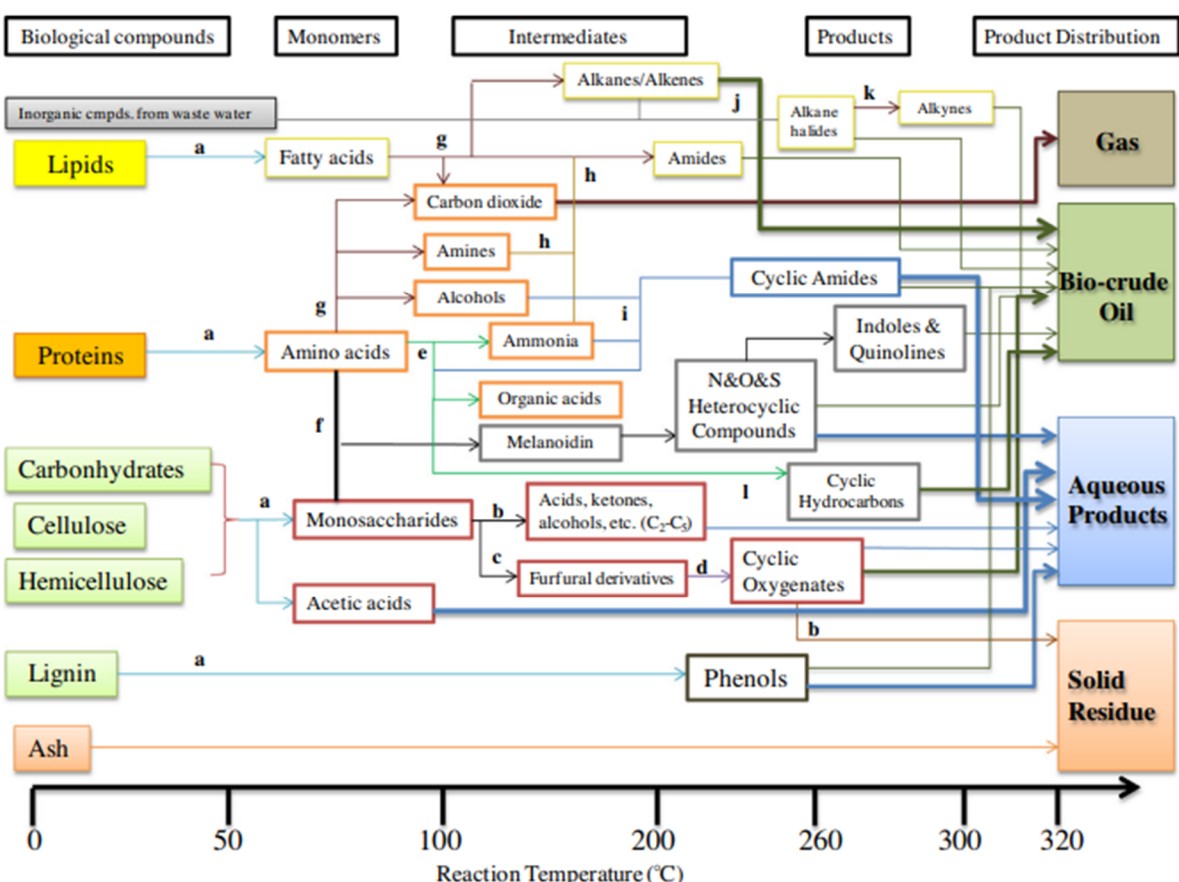

**Figure 4.** Potential reaction pathways for HTL of AW: (a) hydrolysis; (b) decomposition; (c) dehydration; (d) polymerization; (e) deamination; (f) Maillard reaction; (g) decarboxylation; (h) Aminolysis; (i) cyclization; (j) halogenations; (k) dehydrohalogenation; (l) condensation + pyrolysis. Adopted from [81].

It takes at least 15 min for the reaction for protein-containing biomass, such as swine dung, to produce biocrude compounds resembling asphalt. Algal biomass typically requires 10 min to produce self-separated biocrude products. The reaction kinetics of the HTL process are significantly influenced by reaction time. A maximum biocrude yield can only be achieved with a sufficient reaction period; however, an excessive reaction time will result in the formation of charcoal or gaseous products, lowering the biocrude yield.

During the HTL, additional processes, including condensation and repolymerization, could impact the quality of the biocrude. The effectiveness of the HTL reaction and the conversion of the biocrude could both be considerably impacted by heating rates. Faeth et al. conducted a fast HTL conversion of *Nannochloropsis* (heating rates as high as 230 °C/min). They reported that an optimal biocrude yield (66 wt.% based on dry ash-free biomass) can be obtained in less than one minute. Biocrude has a comparable carbon

content and HHV to those converted from conventional HTL. By using a continuous HTL reactor at 250–350 °C, Jazrawi et al. achieved the greatest algal biocrude output (41.7 wt.%) in 3–5 min. In many HTL tests, the reactor head space was pumped with gas before heating up to create an initial pressure. Applying initial pressure primarily serves to keep water in the liquid phase, lower the enthalpy of water's phase change, increase the solubility of biomass, and increase energy efficiency. He et al. [125] observed that no biocrude was generated until the initial pressure reached 0.69 MPa when using 0–1.30 MPa of nitrogen gas ($N_2$) as the initial pressure to convert swine dung into biocrude via HTL. However, numerous types of research have shown that further boosting the initial pressure will not increase the output of biocrude. Zhang et al. [126] increased the initial pressure to convert *Chlorella* into biocrude from 0.69 to 3.45 MPa [127]. It was revealed that beginning pressures above 0.69 MPa had no discernible impact on the yields of HTL products. Yu et al. [128] demonstrated that the HTL product yields converted from *Chlorella* were not significantly affected by raising the starting pressure from 0 to 0.69 MPa.

On the other hand, Yin et al. transformed cattle dung using an initial pressure of 0–0.69 MPa of CO and $N_2$. When the starting pressure was raised, a declining biocrude output was seen, possibly as a result of the self-condensation reaction that turns the biocrude into solid residues. There are no conclusive findings regarding the impact of the starting pressure on biocrude yields while converting protein-containing biomass via HTL [129].

To stabilize the fragmented products of HTL, reduce free radical condensation, cyclization, and repolymerization, and/or suppress char production, reducing gases have been utilized as the processing gas. The following equations illustrate how $H_2$ can stabilize aromatic radicals (Ar) to produce liquid biocrude products. Using HTL and reducing gases such as $CO_2$ and $H_2$, Yin et al. [130] have turned cow dung into biocrude [39]. They concluded that increasing the biocrude output by 5–15 weight percent can be accomplished by employing CO and $H_2$ as HTL processing gases. He et al. reported similar results when converting swine manure into biocrude by HTL with CO and $H_2$ as processing gases. Despite being expensive and risky alternatives, $H_2$ and CO are excellent at stabilizing fragmented liquefaction products. Synthetic gas ($H_2$/CO), an alternative processing gas, could be another choice to accomplish the same goal [131]. As HTL processing gases, nitrogen, air, and carbon dioxide have all been used. However, other than maintaining reactor headspace pressure and preventing feedstock from gasifying, the function of inert gases in HTL is yet unknown. He et al. showed that utilizing compressed air as the processing gas can achieve a similar biocrude production as those processed with $CO_2$ and $N_2$; however, Yin et al. observed that using air as the HTL processing gas led to a substantially lower biocrude yield than those converted with $N_2$. The relative mass ratio of the processing gas to feedstock was one evident factor in the difference. If the starting gas is too airy, the feedstock can oxidize rather than be transformed into biocrude. One of the crucial factors impacting the output and quality of biocrude is the feedstock's total solids (TS) level. The types of biomass and the technologies used to gather it have a significant impact on the TS of the feed supply [132]. In contrast, swine dung collected from a flushing system with settling normally has a TS of 5–10 wt.%, and that from a solid floor typically has a TS of 20–30 wt. %. Most research used feedstocks with a TS of 10–30 wt.% to obtain suitable energy and financial returns. An excessively high TS could result in issues such as inadequate heat transmission and material management, such as pumps, while more water would result in higher costs for things such as heating and wastewater treatment. According to Jena et al. [133], the HTL of *Spirulina* is best achieved with 20 wt.% of TS, and higher TS levels have no discernible impact on product yields. When employing *Chlorella* as an HTL feedstock, Zhang et al. and Gai et al. reported similar findings [134]. Significantly low biocrude yields (10 wt.%) were obtained when TS was lower than 15 wt.%. The authors hypothesized that in order to amass organic clusters that can be transformed into biocrude, a micro-organic phase may be required as the medium.

### 4.4. Catalysis

Under HTL procedures, homogeneous and heterogeneous catalysts have been studied. The various catalysts utilized for HTL of biomass-containing proteins are summarized in Table 5. The catalyst used in HTL did not always produce favorable results. Using heterogeneous catalysts (HZSM-5 and Raney Ni), Zhang et al. transformed *Chlorella* into biocrude under supercritical ethanol (240–300 °C) without increasing the yield of the biocrude. Similar outcomes were obtained using the catalysts $Na_2CO_3$, Co/Mo, $CoMo/Al_2O_3$, Ni/Al, $Ni/SiO_2$-Al2O3, Pt/Al, Pd/C, Ru/C, and zeolites to convert microalgae and animal dung into biocrude via HTL [135]. The interaction between liquid, solid, gas, and sub- or supercritical phases in HTL of protein-containing biomass and heterogeneous catalysts is very complex, and as a result, catalyst deactivations and severe intraparticle diffusion limitations may also contribute to the ineffectiveness of catalysts. The biocrude yield in the absence of a catalyst was already close to the upper limit of what is achievable given the restriction of the mass balance and the availability of carbon and hydrogen, which is another explanation for the catalysts' limited impact on the HTL biocrude yields. It was shown that the HTL biocrude recovered >80% of the C and H atoms from the algal feedstocks without the need for catalysts. The catalyst loadings have an impact on the HTL process. Anastasakis and Ross [136] employed KOH in the range of 5 to 100 weight percent to investigate how catalyst loadings affected the HTL of macroalgae [131]. As catalyst quantities rose from 5 to 100 wt.%, it was noted that the HTL biocrude yields declined by about 10 wt.% while the heating value of biocrude increased by 1.3 MJ/kg [48]. According to Theegala et al., [137] increasing the catalysts' ($Na_2CO_3$) loading from 5 to 20 wt.% did not increase the output of biocrude produced by HTL's conversion of animal manure. Similar outcomes were also shown when utilizing CaO to convert olive seeds, where the HTL biocrude output declined as the amount of catalysts rose from 5 to 40 wt.%. The HTL biocrude yield appears to benefit from homogeneous catalysts, yet recovering homogeneous catalysts is still difficult. Yu et al. discovered that the algal biocrude output was increased by 5–10 wt.% when they studied the influence of NaOH and $Na_2CO_3$ on the HTL of microalgae at 280 °C. Using HTL, Jena et al. [133] found that adding $Na_2CO_3$ can increase the output of biocrude by around 7 weight percent. To increase the technoeconomic viability of catalytic HTL conversion of wet biomass, catalyst regeneration has been proposed. Ong [138] has employed NaOH to regenerate catalysts such as Raney Ni when processing used newspapers hydrothermally. Because catalysts are frequently expensive and their manufacture might adversely influence the environment, effective and sustainable techniques to regenerate catalysts for HTL processes are urgently needed. To have a broader look, Table 5 presents changes by different process parameters, especially temperature and catalyst on biocrude properties.

**Table 5.** Effect of temperature and catalyst on biocrude yield and HHV.

| Feedstocks | Temp (°C) | Catalyst | Non-Cat-Yield (%) | Cat-Yield (%) | Change in Yield (%), by Value | Change in C (%) by Value | Change in N (%) by Value | Change in HHV (MJ/kg) | Ref. |
|---|---|---|---|---|---|---|---|---|---|
| **Lignocellulosic biomass** | | | | | | | | | |
| Wood (birchwood sawdust) | 300 | $K_2CO_3$ | 19.11 | 38 | 19 | −4.00 | −0.03 | −3.00 | [139] |
| Wood (birchwood sawdust) | 300 | KOH | 19.11 | 39 | 20 | −5.00 | 0.00 | −3.00 | [139] |
| Wood (birchwood sawdust) | 300 | $FeSO_4$ | 19.11 | 32 | 13 | −4.00 | 0.03 | 2.00 | [139] |
| Wood (birchwood sawdust) | 300 | MgO | 19.11 | 30 | 11 | −5.00 | 0.03 | −2.00 | [139] |
| Oak Wood | 330 | Nickel Powder | 33.12 | 35 | 2 | −1.00 | 0.00 | 0.31 | [140] |
| Eucalyptus | 350 | $K_2CO_3$ | 33.12 | 37 | 4 | 3.53 | −0.09 | 1.54 | [141] |
| Eucalyptus | 400 | $K_2CO_3$ | 27 | 29 | 2 | −6.50 | −0.15 | 2.93 | [141] |
| Wheat straw | 350 | $K_2CO_3$ | 26 | 32 | 6 | 2.61 | −0.19 | 0.54 | [142] |
| Wheat straw | 400 | $K_2CO_3$ | 24 | 23 | −1 | −1.18 | 0.27 | −0.47 | [142] |
| Barley straw | 300 | $K_2CO_3$ | 18 | 34 | 16 | 5.26 | 0.06 | 2.42 | [99] |

**Table 5.** *Cont.*

| Feedstocks | Temp (°C) | Catalyst | Non-Cat-Yield (%) | Cat-Yield (%) | Change in Yield (%), by Value | Change in C (%) by Value | Change in N (%) by Value | Change in HHV (MJ/kg) | Ref. |
|---|---|---|---|---|---|---|---|---|---|
| | | | | **Animal Manures** | | | | | |
| Cow manure | 350 | $K_2CO_3$ | 41 | 35 | −6 | 9.00 | 0.90 | 3.00 | [142] |
| Cow manure | 400 | $K_2CO_3$ | 32.37 | 32.29 | −0.01 | 4.41 | 0.00 | 0.00 | [142] |
| Swine manure | 350 | $K_2CO_3$ | 41 | 37 | −4 | 5.00 | 0.60 | 3.30 | [142] |
| Swine manure | 400 | $K_2CO_3$ | 36.97 | 34.76 | −2 | −0.33 | −0.09 | 3.09 | [142] |
| | | | | **Protein-containing biomass** | | | | | |
| Fish sludge | 350 | $K_2CO_3$ | 59 | 51 | −8 | 0.92 | 0.17 | 0.50 | [142] |
| Fish sludge | 400 | $K_2CO_3$ | 51.27 | 47.17 | −4 | 0.85 | 0.35 | 0.12 | [142] |
| Sewage sludge | 350 | $K_2CO_3$ | 40.65 | 45 | 4.45 | 2.51 | −1.02 | 1.30 | [12] |
| Sewage sludge | 400 | $K_2CO_3$ | 40.13 | 43 | 2.87 | 1.15 | −1.43 | 0.26 | [12] |
| Sewage sludge | 300 | $NiMo/Al_2O_3$ | 27 | 24 | −3 | 4 | 0.03 | 2.42 | [143] |
| Sewage sludge | 300 | $CoMo/Al_2O_3$ | 27 | 21 | −6 | 1 | 2.29 | 0.42 | [143] |
| Sewage sludge | 300 | Activated Carbon | 27 | 23 | −4 | 1 | 1.30 | 3.42 | [143] |
| Biopulp (Food waste) | 350 | $K_2CO_3$ | 28.9 | 36.6 | 7.5 | 1.24 | −0.09 | 0.80 | [144] |
| Spent compost mushroom | 400 | $K_2CO_3$ | 22.86 | 20.42 | −1.8 | −1.47 | −0.65 | −0.67 | [145] |
| Macroalgae (*Ulva prolifera*) | 280 | MgO | 17 | 16 | −1 | 13 | −0.4 | −2.80 | [146] |
| Macroalgae (*Ulva prolifera*) | 280 | $Al_2O_3$ | 17 | 26 | 9 | 10 | −0.7 | −3.40 | [146] |
| Macroalgae (*Ulva prolifera*) | 280 | $MgCl_2$ | 17 | 27 | 10 | 11 | −0.2 | 0.60 | [146] |
| Microalgae (*Chlorella vulgaris*) | 350 | $Na_2CO_3$ | 38 | 23 | −15 | 2.90 | −1.60 | 2.00 | [142] |
| Microalgae (*Nannochloropsis*) | 350 | $Na_2CO_3$ | 37 | 21 | −16 | 1.50 | −0.30 | 1.00 | [142] |
| Microalgae (*Porphyridium*) | 350 | $Na_2CO_3$ | 21 | 21 | 0.00 | −26.70 | −2.20 | −12.90 | [142] |
| Microalgae (*Nannochloropsis*) | 350 | Pd/C | 35 | 57 | 22 | −2 | −0.38 | 0.00 | [147] |
| Microalgae (*Chlorella vulgaris*) | 300 | $NiMo/Al_2O_3$ | 32 | 29 | −3 | 10.77 | 3.52 | 4.10 | [143] |
| Microalgae (*Chlorella vulgaris*) | 300 | $CoMo/Al_2O_3$ | 32 | 35 | 3 | 12.19 | 3.49 | 4.97 | [143] |
| Soyabean oil (Triyglycerides) | 320 | $KH_2PO_4$ | 85.9 | 93.5 | 8.85 | 1.40 | 0.22 | −1.00 | [148] |
| Soy protein | 320 | $K_2HPO_4$ | 28.85 | 31.25 | 2.95 | 0.54 | 0.39 | −0.14 | [149] |
| Potato starch | 320 | $K_2HPO_4$ | 11.07 | 22.1 | 11 | 2.60 | −0.01 | 3.04 | [148] |
| Potato starch | 320 | $Na_2CO_3$ | 11.07 | 18.77 | 7.6 | −1.32 | 0.00 | 2.14 | [148] |
| Human feces | 330 | $Ni-Tm/TiO_2$ | 40 | 44 | 4 | −4.8 | 1.18 | −4.7 | [149] |
| Human feces | 330 | $Tm/TiO_2$ | 40 | 40 | 0.0 | −2.6 | 0.47 | −2.7 | [149] |
| Human feces | 330 | $Ni/TiO_2$ | 40 | 41 | 1 | −3.8 | 0.64 | −2.7 | [149] |
| Human feces | 330 | $TiO_2$ | 40 | 40 | 0.0 | 0.79 | 0.79 | −0.78 | [149] |

*4.5. HTL Products Separation*

After the HTL reaction, the HTL products can be divided into four categories: biocrude, solid residue, aqueous products, and gaseous products. A typical separation process for HTL products depends upon the type of feedstock, process mode, and process conditions. At batch scale, the biocrude fraction from HTL products was extracted using organic solvents such as dichloromethane, acetone, and toluene. The effects of various organic solvents on the yields and quality of HTL products have been researched by Valdez et al. [138]. The biocrude produced by *Nannochloropsis* was recovered using non-polar solvents (hexadecane, decane, hexane, and cyclohexane) and polar solvents (methoxycyclopentane, chloroform, and dichloromethane). It has been discovered that non-polar solvents, including hexadecane and decane, can produce high gravimetric yields of biocrude (39 wt.%). Still, the recovered biocrude has a lower carbon content than that recovered with polar solvents, such as dichloromethane (69 wt.% for decane) (76 wt.%). The solvent used significantly impacts the amount of free fatty acids recovered from the biocrude, with polar solvents recovering more fatty acids than non-polar solvents.

Water and toluene were used sequentially to extract HTL biocrude from wet biowaste and remove nitrogen-containing chemicals from the HTL biocrude. The nitrogen content of algal biocrude was reduced by 16% when an ultrasonically aided water extraction was carried out, but the carbon and hydrogen contents increased. However, alternative extraction methods were also recommended to be considered to increase the effectiveness of extractive denitrogenation and decrease the amount of biocrude lost into the water extract for HTL biocrude. Although HTL biocrude is often recovered using solvents, certain research, especially those concentrating on developing continuous HTL reactors, separate HTL biocrude by decanting the solid products from aqueous products. For instance, no organic solvent was used during the continuous hydrothermal treatment of microalgae in the Pacific Northwest National Laboratory (PNNL) investigations to recover the algal biocrude. Continuous HTL of swine manure uses a similar separating technique.

## 5. Challenges in HTL and Recommendations

This review article addresses the latest trends of HTL of lignocellulosic and protein-containing biomass, covering their pretreatment methods, chemical reaction mechanisms, process mode, and process conditions, which affect the productivity of biocrude and overall efficiency of HTL processing. After reviewing numerous studies, it has been concluded that HTL is suitable for treating any type of biomass. However, HTL still is at batch or pilot scale due to some gaps that need to be filled for its technological advancement.

### 5.1. Feedstocks Perspective

Undoubtedly, lignocellulosic biomass has been an old source for HTL; however, there are some demerits such as biomass cost, deforestation, and slurry pumpability in HTL. The cost and deforestation can be tackled by selecting lignocellulosic waste biomass such as wheat straw, rice husk, etc. Further pumpability can be enhanced by adding substrates such as glycerol, etc., or co-liquefaction with other biomass like microalgae. Another drawback of the lignocellulosic biomass is that they are dry and require fresh water for the slurry preparation. Over the years, researchers have tried to recirculate the aqueous phase (water generated from the HTL process) for saving freshwater cost and biocrude yield enhancement, however studies reported that recirculating aqueous phase approach imparts higher nitrogen in biocrude with successive rounds of recycling [4,111,112]. High protein-holding biomass such as microalgae, sewage sludge, and food waste are often considered prone to high biocrude productivity; on the other side, these provide higher nitrogen in biocrude, which increases the cost of hydrotreatment and reduces the catalyst life. Therefore, further technoeconomic studies are required to calculate the energy cost between biocrude productivity through HTL and removal of nitrogen via hydrotreatment, including the catalyst cost.

### 5.2. Setting a Common Paradigm for Product Processing

To develop a reliable HTL process, the performance parameters should be well defined in a unique way, such as the HHV of both feedstock and biocrude, as well as the yields of biocrude on a dry ash-free basis. It also recommended that a unique model for processing HTL products should be developed. At batch scale, the usage of solvents (such as acetone, ethanol, diethyl ether DEE) is very common and has been practiced over the years. This could increase the cost of the HTL process and can alter the biocrude yield and quality for large-scale plants. For both pilot and commercial facilities, gravimetric separation is more favorable and cost-effective.

### 5.3. Process Mode

By reviewing several studies on HTL continuous plants, it is recommended that if plant reactor configuration is tubular, it has some benefits such as no moving parts, ease of scalability and if reactor combinations (CSTR, PFR) has also an advantage in reducing plugging, allowing faster heat exchange. The heat transfer can also be enhanced by setting

oscillating flow. In HTL, continuous process of highly concentrated slurries requires stable pumping, which needs technoeconomic viability of the process. The reactor material is a key concern; hence, selecting the material requires the utmost attention according to the reaction environment. While selecting reactor material, plant operating life and cost must be sensibly assessed.

### 5.4. Undertaking the Whole Chain (HTL with Catalytic Hydrotreatment)

The research scope must be enhanced to biocrude upgradation such as deoxygenation and denitrogenation instead of only liquefying biomass into biocrude to achieve the goal. The inorganics in biocrude, mainly metals, must be focused on as these can directly impact the design and viability of the upgrading stage due to catalysts. It is important to quantify the $H_2$ consumption when upgrading the process, especially when hydrogen is supplied by external sources, which is cost-effective to make the system sustainable; hydrogen must be produced from the HTL process and renewable energy sources such as solar or wind. Different strategies can be followed for upgrading biocrude such as co-processing biocrude, fossil crude fractions in existing hydro treaters, co-processing in fluid catalytic crackers, fractional distillation of biocrude, followed by co-processing of the single fractional cuts.

**Author Contributions:** A.S.J. (Conceptualization, investigation, data curation, writing—original draft preparation, writing—review and editing). A.A.S. (Conceptualization, investigation, writing—review and editing, visualization, and supervision). J.A. (Investigation and data curation). S.R. (Investigation and data curation). S.H.S., A.K.S., A.R., N.M.M., Z.H., M.A.U. and M.M. (All contributing to writing—original draft preparation, writing—review and editing). All authors have read and agreed to the published version of the manuscript.

**Funding:** This research received no external funding.

**Conflicts of Interest:** The authors declare no conflict of interest.

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
