# Peer review of "Hydrothermal Liquefaction of Lignocellulosic and Protein-Containing Biomass: A Comprehensive Review"

_catalysts, doi:10.3390/catal12121621_

Round 1

Reviewer 1 Report

This work reports on covers the Hydrothermal liquefaction of various feedstocks especially lignocellulosic, especially the information of the chemical reaction mechanisms of the main components in the HTL, and primary parameters that influence the quantity and characteristics of HTL products. The review paper addresses key trends in the use of HTL, and presents challenges and recommendations for the process. The process has been explored and described in detail, manuscript is well structured and organized. After careful consideration, I recommend accept in present form.

Author Response

Reviewer 1

This work reports on covers the Hydrothermal liquefaction of various feedstocks especially lignocellulosic, especially the information of the chemical reaction mechanisms of the main components in the HTL, and primary parameters that influence the quantity and characteristics of HTL products. The review paper addresses key trends in the use of HTL, and presents challenges and recommendations for the process. The process has been explored and described in detail, manuscript is well structured and organized. After careful consideration, I recommend accept in present form.

Response Thanks for the appreciation and acceptance of the paper.

Reviewer 2 Report

The authors have down an extensive study of the HTL method with appropriate number of references.

The paper is well structured. There are some typo mistakes and wrongly numbered Tables and Figures;

Page 13 Table 1, Page 16 to 17 the indexes of the molecules needs to be corrected.

Author Response

Reviewer 2

Q1 The authors have down an extensive study of the HTL method with appropriate number of references.

Response Thanks for the valuable comments

Q2 The paper is well structured. There are some typo mistakes and wrongly numbered Tables and Figures;

Response thanks for the valuable comments based on suggestion correction has been made in overall manuscript.

Page 13 Table 1, Page 16 to 17 the indexes of the molecules needs to be corrected.

Response Author thanks for the valuable suggestion correction has been made as per suggestion

Reviewer 3 Report

GENERAL OBSERVATIONS AND ISSUES

The review is very comprehensive and the data is presented logically and with reasonably good organization.  I like the report and think it will be very useful to the waste conversion field in general as well as the HTL field in particular.  Overall this is a very good review, however there are some technical issues that must be corrected before publication.

1.     The references are all mixed up after about 95 to 108.  Some appear to be missing.  Several authors are cited in the paper but do not have a corresponding listing in the References section.  I have noted some of these in my detailed list below, however I am sure I did not catch all of them

2.     Many chemical formulae need to be corrected to put the numbers in subscript

3.     Capitalization is mixed up in many of the table entries and in some of the text

4.     In general, the tables are spaced poorly and difficult to read

SPECIFIC ISSUES

141- b-1,4

160- (180 oC)

It would be nice to have the reaction products formed/bonds broken or formed from each of the major lignocellulose constituents illustrated graphically as a function of temperature, pH, and/or pressure.  Similarly, the impact of catalysts in product shifting would make a good table.

192- I do not understand the reference to “organic carbon”

217- Are the authors referring to lignin “oligomers, dimers, and monomers”? How do they act as catalysts?

229- Need to define “coke” as this is typically formed from coal or sometimes petroleum oil

256- Nagel and Zhang not referenced

264- 240 oC.

Table 2- “More residence time”, “Can be carried out at...”

Table 2- the Comments column is very difficult to align with the row each comment is supposed to address

Table 2- I do not see the “a, b, c” footnotes indicated for LHW, AFEX, ARP

Table 2- Concentrated acid and Diluted acid rows should be “inhibitor”, not “inhabitor”

Table 3- spacing is messed up

Table 3- first line in comments- what is “In-shortage”?

286- lipid

341- Botryococcus braunii

407- “Studies have shown that lignocelluloses with higher cellulose and hemicellulose concentrations produce more bio-oil than those with lower lignin contents”.  This is confusing.  Almost by definition, higher cellulose and hemicellulose means there is lower lignin levels. 

412- Feng et al. (2014) not referenced

416- “However, because lignin has a better thermal stability than the other lignocellulose components, it can break-down at higher temperatures” is not correct.  Something like “Due to lignin’s higher thermal stability, its breakdown requires higher temperatures than cellulose and hemicellulose”.

420- Chan (2015) not referenced

420- “As phenolic compounds made up the majority of bio-oil (more than 72.86 percent), lignin predominated in the process of producing bio-oil through high temperature breakdown” is seemingly contradictory to the statement above that higher cellulose and hemicellose content results in more bio-oil.

Figure 2- spelling errors- Trunks not truncks and Trees not tress

428- 4.2 Process mode-  The first paragraph in this section makes no sense.  It appears that the authors are trying to compare algal and swine dung production of bio-oil, but the comparisons are not clear.   

436-439- These two sentences are not connected.  How does the swine faeces processing impact the algal results?

450- [99]. 

500- remove period at start

541- He is not referenced

560- Jena is not referenced

571- Zhang does not exist in reference list

584- How do you get to 100% KOH catalyst weight percent?  There would be no feedstock

585- There is no Anastasakis and Ross reference

588- Theegala does not exist in reference list

594- Yu is not referenced

596- Jena is not referenced

599- Ong is not referenced

612- “…HTL products is.”  This is an incomplete sentence

616- Valdez not referenced

Author Response

Reviewer 3

GENERAL OBSERVATIONS AND ISSUES

The review is very comprehensive and the data is presented logically and with reasonably good organization.  I like the report and think it will be very useful to the waste conversion field in general as well as the HTL field in particular.  Overall this is a very good review, however there are some technical issues that must be corrected before publication.

  1.     The references are all mixed up after about 95 to 108.  Some appear to be missing.  Several authors are cited in the paper but do not have a corresponding listing in the References section.  I have noted some of these in my detailed list below, however I am sure I did not catch all of them

Response: Thanks for the valuable suggestion, correction has been made related to reference from 95-108.

  1.     Many chemical formulae need to be corrected to put the numbers in subscript

Response: The author thanks for the valuable suggestion correction has been made in chemical formulae.

  1.     Capitalization is mixed up in many of the table entries and in some of the text

Response: Thanks for the valuable suggestion capitalization is now set as per suggestion

  1.     In general, the tables are spaced poorly and difficult to read. 

Thanks for the valuable suggestion based on suggestion correction has been made in all tables

SPECIFIC ISSUES

141- b-1,4

160- (180 oC)

Response: Correction has been made as per suggestion

It would be nice to have the reaction products formed/bonds broken or formed from each of the major lignocellulose constituents illustrated graphically as a function of temperature, pH, and/or pressure.  Similarly, the impact of catalysts in product shifting would make a good table.

Response:

Response: Yes, it’s a good suggestion, the studies related to pressure, pH verses the bond cleavage are not too much available in the literature, however, there are some studies which could describe the relationship of temperature verses the breaking of bonds of different chemical constitutions of biomass, specially carbohydrates, proteins, lignin, and etc. the figure 4, with its reference is attached (incorporated in manuscript now, in section 4.3). We have already made a table 5, where we described the effect of catalyst on HTL products. However, the catalyst & pH vs cleavage of bonds would be a good gap for future working.

192- I do not understand the reference to “organic carbon”

Response: Correction has been made as per suggestion

217- Are the authors referring to lignin “oligomers, dimers, and monomers”? How do they act as catalysts?

Response: Yes lignin can be used as a catalyst  “typically, the lignin-to-catalyst transformation involves acid treatment, thermal treatment and surface modification”

Zhu, Y.; Li, Z.; Chen, J. Applications of lignin-derived catalysts for green synthesis. Green Energy & Environment 2019, 4, 210-244, doi:https://doi.org/10.1016/j.gee.2019.01.003.

229- Need to define “coke” as this is typically formed from coal or sometimes petroleum oil

256- Nagel and Zhang not referenced

Reference has been added

264- 240 oC.

Ye it has been corrected

Table 2- “More residence time”, “Can be carried out at...”

Table 2- the Comments column is very difficult to align with the row each comment is supposed to address

Table 2- I do not see the “a, b, c” footnotes indicated for LHW, AFEX, ARP

Table 2- Concentrated acid and Diluted acid rows should be “inhibitor”, not “inhabitor”

Table 3- spacing is messed up

Table 3- first line in comments- what is “In-shortage”?

Response Correction has been made in table 2 and 3 as per suggestion

286- lipid

341- Botryococcus braunii

407- “Studies have shown that lignocelluloses with higher cellulose and hemicellulose concentrations produce more bio-oil than those with lower lignin contents”.  This is confusing.  Almost by definition, higher cellulose and hemicellulose means there is lower lignin levels.

Response: Thanks for the valuable suggestion correction has been made

412- Feng et al. (2014) not referenced

Response: Have now corrected

416- “However, because lignin has a better thermal stability than the other lignocellulose components, it can break-down at higher temperatures” is not correct.  Something like “Due to lignin’s higher thermal stability, its breakdown requires higher temperatures than cellulose and hemicellulose”.

Response: Had now corrected as per suggestion

420- Chan (2015) not referenced

Response: Now reference has been added

420- “As phenolic compounds made up the majority of bio-oil (more than 72.86 percent), lignin predominated in the process of producing bio-oil through high temperature breakdown” is seemingly contradictory to the statement above that higher cellulose and hemicellose content results in more bio-oil.

Yes, you rightly pointed, this seems to be out of context statement. Now we have revised this paragraph by putting relevant information with logical reasoning, as incorporated in manuscript with few more references in section 4.1.

Statements added:

Lignin is thermally more stable than the other biomass, and the order of hydrothermal conversion degree of biomass and biomass component was as follows: cellulose, sawdust, rice husk, and then lignin. In the liquefaction of switchgrass in subcritical water, the residue solid mainly contained lignin fractions.

The soft wood biomass contains higher lignin content than hard wood biomass, it was reported the lignin rich cypress (soft wood biomass) produced the hydrocarbons with major portion of phenolic hydrocarbons and derivatives than cherry (hard wood biomass).

In the hydrothermal conversion of the mixtures from different ratio of cellulose to lignin, the char yields increased with the increasing lignin content, and the yields of gas and aqueous soluble products increased with the increasing cellulose content, but it was difficult to conclude the oil yields change with increasing lignin content.

Reference A: Hydrothermal conversion of lignin: A review

Figure 2- spelling errors- Trunks not truncks and Trees not tress

Response Thanks for the valuable suggestion correction has been made

428- 4.2 Process mode-  The first paragraph in this section makes no sense.  It appears that the authors are trying to compare algal and swine dung production of bio-oil, but the comparisons are not clear.  

Response Thanks for the valuable suggestion and correction, based on suggestion new paragraphs has been added

436-439- These two sentences are not connected.  How does the swine faeces processing impact the algal results?

Response Correction has been made

450- [99].

500- remove period at start

Response Correction has been made

541- He is not referenced

Response Correction has been made

560- Jena is not referenced

Response Correction has been made

571- Zhang does not exist in reference list

Response Correction has been made

584- How do you get to 100% KOH catalyst weight percent?  There would be no feedstock

Response Correction has been made

585- There is no Anastasakis and Ross reference

Response Correction has been made

588- Theegala does not exist in reference list

Response Correction has been made

594- Yu is not referenced

Response Correction has been made

596- Jena is not referenced

Response Correction has been made

599- Ong is not referenced

Response Correction has been made

612- “…HTL products is.”  This is an incomplete sentence

Response Correction has been made

616- Valdez not referenced. 

Response Correction has been made

Reviewer 4 Report

The manuscript interestingly present a complete review of biomass hydrotreatment. The manuscript is well written, organized and in the scope of the special issue. There are some points that need attention before it can be published:

·       Please carry an English revision looking for typo mistakes. There are also superscript and subscribed letters that need correction, such as in molecular formulas.

·       Line 141: please add “beta” on “-1, 4-glycosidic”

·       Please be careful with too long paragraphs such as in lines 212-269 and 284-376. Please look up the entire manuscript.

·       Table 2: acid pretreatment is divided into diluted and concentrated. Please remove acid pretreatment from the table and left only dilute and concentrated acid pretreatment.

·       Table 5 – please define HHV

·       Line 611: “A typical separation process for HTL products is.” Please complete this sentence.

Author Response

Reviewer 4

Comments and Suggestions for Authors

The manuscript interestingly present a complete review of biomass hydrotreatment. The manuscript is well written, organized and in the scope of the special issue. There are some points that need attention before it can be published:

  •       Please carry an English revision looking for typo mistakes. There are also superscript and subscribed letters that need correction, such as in molecular formulas.

Response Correction has been made via doing careful proofreading

  •       Line 141: please add “beta” on “-1, 4-glycosidic”

  •       Please be careful with too long paragraphs such as in lines 212-269 and 284-376. Please look up the entire manuscript.

Response Thanks for the valuable suggestion correction has been made as per suggestion in lines 212-269 and 284-376

  •       Table 2: acid pretreatment is divided into diluted and concentrated. Please remove acid pretreatment from the table and left only dilute and concentrated acid pretreatment.

Response Thanks for the comments correction has been made

  •       Table 5 – please define HHV

Response thanks for the valuable suggestion HHV is higher heating value

  •       Line 611: “A typical separation process for HTL products is.” Please complete this sentence.

Response thanks for the comments correction has been made in line 762 and onwards.

“A typical separation process for HTL products depending upon the type of feedstock, process mode, and process conditions”.
